# Si–Fe–C–N Coatings for Biomedical Applications: A Combinatorial Approach

**DOI:** 10.3390/ma13092074

**Published:** 2020-04-30

**Authors:** Charlotte Skjöldebrand, Gry Hulsart-Billström, Håkan Engqvist, Cecilia Persson

**Affiliations:** 1Department of Materials Science and Engineering, Faculty of Science and Technology, Uppsala University, 752 37 Uppsala, Sweden; charlotte.skjoldebrand@angstrom.uu.se (C.S.); hakan.engqvist@angstrom.uu.se (H.E.); 2Department of Surgical Sciences, Faculty of Medicine and Pharmacy, Uppsala University, 751 83 Uppsala, Sweden; gry.hulsart@surgsci.uu.se

**Keywords:** coating, silicon nitride, iron, carbon, mechanical properties, surface roughness, biocompatibility, implant

## Abstract

Ceramic coatings may prolong the lifetime of joint implants. Certain ions and wear debris may however lead to negative biological effects. SiN-based materials may substantially reduce these effects, but still need optimization for the application. In this study, a combinatorial deposition method enabled an efficient evaluation of a range of Si–Fe–C–N coating compositions on the same sample. The results revealed compositional gradients of Si (26.0–33.9 at.%), Fe (9.6–20.9 at.%), C (8.2–13.9 at.%) and N (39.7–47.2 at.%), and low oxygen contaminations (0.3–0.6 at.%). The mechanical properties varied with a hardness (H) ranging between 13.7–17.3 GPa and an indentation modulus (M) between 190–212 GPa. Both H and M correlated with the Si (H and M increased as Si increased) and Fe (H and M decreased as Fe increased) content. A slightly columnar morphology was observed in cross-sections, as well as a surface roughness in the nm range. A cell study revealed adhering pre-osteogenic MC3T3 cells, with a morphology similar to that of cells seeded on a tissue culture plastic control. The investigated coatings could be considered for further investigation due to the ability to tune their mechanical properties while maintaining a smooth surface, together with their promising in vitro cell response.

## 1. Introduction

The demand for hip and knee replacement surgery is predicted to increase due to both increasing life expectancy as well as an increasing need in younger patients [1,2,3,4]; total hip replacement surgery has increased and is estimated to further increase by 174% [4] in the United States and 134% [2] in England and Wales between 2010 and 2030. The corresponding projections for primary knee replacements are as high as 637% [4] and 117% [2], respectively. As the primary surgeries become more prevalent so do the revision surgeries, with predicted increases as high as 137% and 1264% for total hip arthroplasty revisions and 601% and 332% for total knee arthroplasty revisions in the United States and England plus Wales, respectively. To improve the quality of life for patients undergoing joint replacements it is of utmost importance to increase the replacement’s lifespan in order to minimize the need for revision surgeries, which can be both costly and painful.

The most common materials used in total joint replacement bearing surfaces are a metal, cobalt chromium alloy (CoCr), paired with a polymer, commonly highly crosslinked polyethylene (HXLPE). However, the use of ceramic materials, such as zirconia-toughened alumina (still paired with HXLPE) for hip replacements has been steadily growing [5]. The average implant failure rate is approximately 2% after 10 years [6], with the most common failure mechanism being aseptic loosening [5,6,7,8,9,10,11]. This is believed to be caused, at least in part, by the generation of wear debris. Therefore, a strategy for achieving a longer lifespan of joint implants is to improve their wear properties through the use of ceramic coatings on the metallic parts. This strategy could give a better wear resistance, as well as decrease the corrosion rate of the underlying metal. Potential coating materials investigated include e.g., diamond-like carbon, zirconium nitride and titanium nitride [12,13,14]. Of these, titanium nitride, zirconium nitride and zirconium oxide are currently commercially available. However, these coatings are yet to show an increase in the implant lifespan [15], which could be due to short follow up as well as the use of coated implants still being limited resulting in a small patient group. 

A particularly promising coating material is amorphous silicon nitride [16,17,18,19,20,21,22,23,24,25] as bulk silicon nitride was shown to be biocompatible [26,27,28,29,30] and slowly dissolves in water-based solutions [19,20,31,32,33]. Small, soluble, and biocompatible wear particles have the potential to reduce the negative immune response and prolong the implant lifetime. Previous studies have found that silicon nitride coatings can have a high hardness (up to 26 GPa) and Young’s modulus (up to 212 GPa) [17,23,34], low wear rates, generate biocompatible wear particles [27] and act as a protective barrier to metal release from the underlying metal [19]. Despite the promising properties of silicon nitride coatings, some areas of concern still remain, such as difficulties in achieving a good adhesion [17] and optimization of the dissolution rate [19]. These are, at least in part, related to the mechanical properties and microstructure of the coatings and could therefore be altered by alloying the coating with other elements. The addition of carbon to silicon nitride coatings has for example been shown to affect the bonding structure through the introduction of Si–C bonds [21,25]. The change in bonding structure could also alter the dissolution rate due to a difference in bonding strength between Si–N and Si–C. In addition, the hardness, which is important to optimize for coating adhesion [35,36], correlates to the composition [16,18,24]. Introducing a metal, such as iron, to the system may influence the mechanical properties and furthermore, these metal-containing systems have a higher deposition rate than semi-conductors due to their conductivity. Iron ions are also generally well accepted by the body [37,38,39,40].

An efficient way to investigate the influence of the elemental composition on mechanical properties and structure is to use compositional gradients. This method is known as combinatorial sputtering and has the potential to provide information on a wide range of compositions using only a few samples [41,42]. This is achieved by generating a gradient which allows several compositions to be investigated on the same sample, and thus allows for an efficient use of both material and time. Therefore, this study aimed to investigate the composition, mechanical properties, surface morphology, crystal structure and biocompatibility of silicon nitride coatings incorporating carbon and iron using a combinatorial approach.

## 2. Materials and Methods

### 2.1. Coating Deposition

Coatings were grown on silicon wafers using reactive magnetron sputtering (Figure 1a). The sputtering chamber was built in-house and consisted of a main chamber and load lock. The vacuum in the main chamber was obtained and maintained with a 3001 s^−1^ turbo pump backed by a membrane pump. The sputtering gas, in this case argon, was introduced close to each magnetron (mightyMAK, ScoTech Ltd., Renfrewshire, Scotland), while the reactive nitrogen gas was introduced close to the substrate. The gas flow was controlled by a mass flow controller. The targets were mounted at a distance of 175 mm from the surface and at an angle of 38.81° to the surface normal to achieve compositional gradients. The targets were protected by electronically controlled shutters controlled using LabView.

The base pressure in the chamber before deposition was 10^−9^ Torr (10^−7^ Pa) and the pressure during deposition was 3 mTorr (400 mPa). The gas flows were 3 sccm (standard cubic centimeter per minute) and 10 sccm respectively, giving a nitrogen-to-argon gas flow ratio ($f_{N_{2}/Ar}$) of 0.323. In a previous study on SiN_x_ coatings [23] a $f_{N_{2}/Ar}$ of 0.3 was found to give high nitrogen content, which was beneficial for the mechanical properties. In another study, a high nitrogen content was found to be beneficial for the dissolution behavior of silicon nitride coatings [19]. The targets, silicon (99.99% pure, ScoTech Ltd), carbon (99.99% pure, ScoTech Ltd) and iron (99.99% pure, Kurt J. Lesker), had a diameter of 7.62 cm (3 in diameter) and were individually powered. To power the silicon target, a pulsed direct current (PDC) with an effect of 200 W, frequency of 200 kHz and on-time of 2 µs was applied. Carbon and iron were powered with direct current (DC) at 65 W and 25 W respectively. During deposition, the substrate was heated to a temperature of 350 °C, which was found to give high hardness and modulus and not reduce the nitrogen content [23]. The deposition time was 10,000 s. The Si-wafer substrates (3 in diameter) were cleaned with ethanol and heat-treated in the deposition chamber for 10 min prior to deposition.

### 2.2. Material Characterization

Due to the placement and angled position of the targets, no two points on the same sample were expected to be alike. Three samples for each analysis and five points on each sample were investigated. The selected points were each corner of a square with a side of 40 mm (point 1–point 4) and a middle point (point 5) [43]. Each point was placed in a Cartesian coordinate system and point 3 was chosen as the origin (Figure 1b) for the later analysis.

#### 2.2.1. Composition

Chemical analysis was performed with time of flight elastic recoil analysis (ToF-ERDA). The accelerator used was a 5MV 15N-SDH2 pelletron (National Electrostatics Corporation, NEC, WI, USA) with a 36 MeV I^8+^ ion beam. The incident angle of the ions was 45° to the sample surface and the detector was located at an angle 22.5° to the sample surface. Systematic uncertainties for light elements such as carbon can be 5–10% of the concentration for absolute measurements [44], while the accuracy for inter-sample comparison is much higher. The CONTES software package, developed at site in Matlab, was used [45].

#### 2.2.2. Crystallinity, Surface, and Cross-Sectional Morphology

The crystalline structure of three samples was assessed using X-ray diffraction (XRD), using a Bragg-Brentano setup and CuKα radiation (D8 Discover, Bruker, Billerica, MA, USA). The 2θ angle spanned from 20° to 80°. The sample was mounted on a parallel stage and held in place with vacuum.

The thickness and morphology of the coatings were evaluated on fractured cross-sections after plasma cleaning in situ in a field-emission gun scanning electron microscopy (SEM) (Merlin, Zeiss, Oberkochen, Germany). The samples were analyzed with an in-lens detector at an acceleration voltage of 3 kV and probe current of 80 pA. 

The surface roughness and topography was analyzed with atomic force microscopy (AFM) (XE150, Park Systems Corp., Suwon, Korea). The analysis was performed with a silicon (N-type) tip with <10 nm tip radius in non-contact mode over an area of 5 µm × 5 µm. 

#### 2.2.3. Hardness and Indentation Modulus

The mechanical properties were assessed using a nanoindenter (UNHT, Anton Paar, Graz, Austria). The indentations were performed on the 5 points of the coatings, as well as an uncoated silicon wafer substrate, using a diamond Berkovich tip (α = 65.3°). Twenty-five indentations to a depth of 50 nm and spaced 5 µm apart were performed at each point (1–5). The procedure was repeated on three samples. The tip was calibrated prior to testing the coated and uncoated silicon wafer and the calibration was confirmed immediately after the test series was complete. Both calibration and calibration confirmations were performed using a certified fused silica calibration sample (Anton Paar TriTech, Peseux, Switzerland). Before testing, samples were cleaned in ethanol. Indentation modulus, M, and hardness, H, were obtained using the Oliver Pharr method [46]. In this study, Poisson’s ratio for the different points was unknown. Therefore, the indentation modulus, without a Poisson’s ratio, was reported, as is common in e.g., the field of heterogeneous crystalline networks [47]. The indentation modulus relates to Young’s modulus by (1 − ν_s_^2^), where ν_s_ is Poisson’s ratio of the sample.

#### 2.2.4. In Vitro Cell Study

The in vitro cell response to the coatings was evaluated using osteoprogenitor (MC3T3) cells. The cells were expanded to 70% confluency in growing flasks and then seeded over the entire sample in a petri dish (Ø 100 mm, Cell culture dishes, VWR®, Radnor, PA, USA) at a density of 10,000 cells/cm^2^. A tissue culture plastic petri dish was used as a positive control. The samples (n = 3) were then incubated for 24 h after which they were stained with Hoechst 33342 (Life Technologies, Thermo Fisher Scientific, Waltham, MA, USA) and CFDA-SE (Sigma-Aldrich, Saint Louis, MO, USA), and imaged in a fluorescent microscope (Axiomager, Zeiss, Oberkochen, Germany) to evaluate the morphology and surface adherence. The in vitro experiment was repeated three times.

#### 2.2.5. Statistical Analysis

Pearson correlations were used in IBM SPSS Statistics v22 to assess any correlations between coating composition and coating properties. In the case of the mechanical properties, hardness and modulus, the values were compared to the average composition of three other samples deposited with the same parameters since it was not possible to perform all investigations on the same samples due to the destructiveness of these tests. Any statistically significant difference between the coatings and the control group in the in vitro cell study was in addition analysed by a 1-way ANOVA followed by Dunnett’s post-hoc test. 

## 3. Results

All data is presented in Table 1. However, for better visualisation and overview the data is also presented as a matrix of scatter plots (Figure 2) and individual graphs of statistically significant correlations are presented in the Appendix A). Furthermore, numerical data is also presented in colour- and bar plots.

The statistical analysis revealed several statistically significant correlations (Figure 2, * in red).

Each square in Figure 2 represents a correlation between an input parameter (found on the x-axis) and an output parameter (found on the y-axis). General trends that could be observed include the correlation between the composition (Si, Fe, N and C) to the positions (x- and y-coordinates), as well as the correlation between hardness, modulus, and thickness to the x-coordinate. Each parameter is presented in more detail below.

### 3.1. Composition

The ToF-ERDA investigation yielded compositional data that revealed clear differences between all five points (Figure 3).

As expected, the silicon content was highest in the points closest to the silicon target during deposition. The middle point, point 5, followed, and the points furthest exhibited the lowest contents. The reverse effect was observed for iron with the highest content in point 2 and 4, while points 1 and 3 had a lower content and point 5 had an intermediate content.

Carbon range from its lowest content in point 1 to the highest in point 4. Based on the positioning of the carbon target during deposition, points 3 and 4 would be expected to have a higher content than 1 and 2. While this is the case, there is a difference between points 1 and 2 and points 3 and 4. This implies that the gradient is not strictly in the y-direction, but also varies in the x-direction. Nitrogen was supplied as a reactive gas during deposition, and therefore it is reasonable to assume that the availability of the gas was uniform across the substrate. The ToF-ERDA results, however, revealed different nitrogen contents in the investigated points. The highest content was found at point 1 and the lowest content at point 4. While the oxygen contaminations were low, both points 2 and 4 exhibited slightly higher oxygen contents compared to points 1 and 2.

### 3.2. Structure and Morphology

The XRD diffractograms (Figure 4) displayed peaks at 69°, attributable to the underlying silicon wafer substrate.

A signal from the substrate was expected due to the very thin nature of the coatings’. Otherwise, the coating did not reveal any peaks indicating there was no long range order and was hence XRD amorphous. However, the cross-section revealed a columnar structure when investigated in the SEM (Figure 5) but the area closest to the substrate displayed a dense amorphous region. 

The columns had slightly different appearances when compared to each other, with points 1 and 3 appearing denser with more narrow columns than points 2 and 4. The columns were also slightly angled, a side effect of the combinatorial set-up during deposition.

The cross-sections were used to determine the thickness, which had its lowest value at point 2 (467 nm) and highest in point 1 (633 nm). The thickness correlated with both the x-coordinate and the silicon content.

Inspection of the surface revealed a smooth morphology with a low surface roughness. A closer examination of the surface revealed a cauliflower appearance (Figure 6) with an average surface roughness of approximately 2 nm for all the investigated points (Table 1). There was no statistically significant correlation between the roughness and any of the other parameters.

### 3.3. Mechanical Properties

The nanoindentation results (Figure 7) revealed clear differences between the five points. The point with the highest hardness and modulus was point 3, point 2 had the lowest hardness and point 4 had the lowest modulus. Both the hardness and modulus correlated with silicon, iron, and nitrogen but not carbon (Figure 2).

### 3.4. Biocompatibility

The MC3T3 cells adhered to the surface of all investigated points (1–5) after 24 h of incubation, with similar morphology to the tissue culture plastic control (Figure 8a–f). The results are further illustrated in a barplot (Figure 8g) and color plot (Figure 8h) displaying the quantified number of cells/mm^2^.

While there was a trend for lower numbers of cells in points 2 and 4, and highest in point 5, no statistically significant differences (*p* = 0.11) were observed in the number of adhered cells between the different points of the sample and the positive control. Cells showed a comparable response to both the sample and control, with cells dividing, adhering and spreading on the sample, in the same manner as on tissue culture plastic.

## 4. Discussion

In this study, SiFeCN coatings were successfully deposited using a combinatorial approach. This co-sputtering results in compositional gradients which can be used to efficiently investigate mechanical and surface properties of different materials. However, thickness gradients are inevitable since the difference in composition is achieved by the target angle and the distance between the target and substrate. This variation in thickness can be a challenge during coating characterization as it can influence e.g., coating residual stresses and adhesion. 

### 4.1. Composition and Thickness

The parameters used to deposit the coatings in this study yielded gradients for all three deposited target materials: silicon, iron, and carbon. The difference in composition was considerable for all elements and the gradients spanned from 26 at.% to 34 at.% for silicon, 10 at.% to 20 at.% for iron and 8 at.% to 14 at.% for carbon. Based on the placement of the targets during deposition, the silicon and iron gradients were assumed to span the x-direction, while the carbon gradient was assumed to be in the y-direction. Indeed, statistical analysis showed statistically significant correlations between silicon and the x-coordinate (*p* < 0.05) and iron and the x-coordinate (*p* < 0.05), while carbon correlated with the y-coordinate (*p* < 0.05). However, when considering the nitrogen content, it ranged from 46 at.% to 40 at.%, despite being supplied as a gas i.e., giving the same availability across the substrate. The nitrogen gradient displayed a statistically significant correlation to silicon (*p* < 0.05), carbon (*p* < 0.05) and iron (*p* < 0.05). While ToF-ERDA results in a reliable composition analysis, it is not possible to determine the bonding and structural environment. Hence ToF-ERDA measurements are not in themselves enough to explain the correlations between the elements. The correlation between nitrogen and silicon (increasing nitrogen with increasing silicon), is likely due to the enthalpy of formation, ΔƒH^Ө^, of Si_3_N_4_ being highly negative at −743 kJ/mol [48]. Normalizing this value to individual atoms produces a value of −247.7 kJmol^−1^atom^−1^, likely lower than any iron nitride since iron nitrides are typically metastable and hard to form. This could also explain why nitrogen seems to decrease with increasing iron content. A similar argument can be made for the correlation of oxygen to iron where iron oxides are highly negative with ΔƒH^Ө^ = −824 kJmol^−1^ for Fe_2_O_3_ and ΔƒH^Ө^ = −1118 kJmol^−1^ for Fe_3_O_4_ (normalized values would be −412 kJmol^−1^atom^−1^ and −373 kJmol^−1^atom^−1^) [48]. If these values are compared to amorphous silicon oxide (220 kJmol^−1^ [49]), the iron oxide values are lower and are thus more likely to form. This may explain the correlation between oxygen and iron (increasing oxygen with increasing iron). The correlation between nitrogen and carbon could be a consequence of the silicon content i.e., when the silicon content increases the nitrogen increases while the carbon decreases. This could indicate that silicon preferentially bonds to nitrogen, which is corroborated through comparison of the enthalpies of formation (ΔƒH^Ө^ = 247.7 kJmol^−1^ for Si–N and ΔƒH^Ө^ = −65 kJmol^−1^ for Si–C). It is hence possible that both nitrogen and carbon correlate with silicon, but not to one another.

Studies suggest it could be preferable to introduce carbon as carbon-containing coatings generally have a lower oxygen content. The introduction of carbon could also decrease the brittleness of the coatings as the bonds would be more covalent due to a lower electronegativity compared to nitrogen (2.55 for carbon and 3.04 for nitrogen on the Pauling scale) [48]. This study did however not provide any statistically significant correlations that would confirm those theories.

As a consequence of the compositional gradients, coating thickness range from 467 nm to 633 nm in the investigated area, resulting in a difference of 166 nm between the thinnest and thickest points. Coating thickness correlate with the silicon content with statistical significance (*p* < 0.05), most likely as a result of silicon’s presence as the dominant element with a higher power density out of the three target elements. The thickness also correlate with the x-coordinate (*p* < 0.05), which may be a result of the placement of the silicon target i.e., the correlation of the thickness to the x-coordinate is a secondary effect.

### 4.2. Crystallinity, Surface and Cross-Section Morphology

There are coating properties that were seemingly unaffected by the composition or coating thickness. For example, the cross-section morphology was similar at all five points, with a dense ‘glassy’ region closest to the substrate and a slightly columnar structure further away from the substrate (Figure 5). The columns were slightly angled which is likely a result of the deposition flow due to the angled targets. The amorphous results correspond to those of a similar SiN_x_ coating, which was characterized as amorphous and nanocrystalline [23,34]. Pettersson et al. [18] found that a comparably low target power of 1 kW yielded a columnar structure very similar to the structure of the coatings in this study. Since the silicon target in this study was powered by a PDC power supply at 200 W (compared to a high power impulse magnetron sputtering aggregate at 1–4 kW used by Petterson et al.), the resulting columnar structure is not surprising as the power is considerably lower. In the work by Johansson and Lewin [50], similar columnar features were observed for a ternary Al-Ge-N system deposited in a similar set-up. XRD analysis revealed the presence of crystalline AlN, with reduced peak intensity with increasing Ge. As the Ge content increased and the crystallinity decreased, they saw fewer distinct columns. This trend resembles the column narrowing found in the Si–N–Fe–C coatings with increasing silicon content. The columnar structure could have negative consequences on the dissolution rate as it might accelerate the dissolution of the coatings and future attempts should be aimed at depositing homogenously dense coatings [51].

Even though the surface roughness remains low and unchanged with a R_a_ value about 2 nm for all five investigated points, the surface has a cauliflower structure which implies a high rate of nucleation. If there are several nucleation sites competing with each other, the coating will form simultaneously in multiple spots and create an irregular surface. The low surface roughness can, in part, be attributed to the substrate temperature during deposition (350 °C). A high substrate temperature during deposition is expected to result in smooth coatings as the energy increases the mobility of the atoms and ions and allows them to find a more energetically favorable position on the surface. When the surface roughness is compared to similar coatings, the R_a_ for the combinatorial Si–N–Fe–C coatings is slightly lower (2 nm compared to 10–15 nm) [23], despite similar deposition parameters. This difference is likely due to the surface roughness of the substrates. The coatings investigated in this study were deposited on Si wafers with a roughness < 1 nm, while previous coatings were mainly deposited on CoCr discs, polished to a R_a_ value of approximately 10 nm [23]. In the study by Pettersson et al. [18], silicon wafers were used as the substrate. They found a surface roughness of 0.2 to 3.8 nm for silicon nitride and silicon carbon nitride coatings.

### 4.3. Mechanical Properties

Special care has to be taken when performing nanoindentation on a coating with varying thicknesses deposited on a silicon wafer in order to minimize the risk of influence from the substrate. This is often accomplished by complying with the commonly used ‘rule of one-tenth’. This rule states that the maximum indentation depth should not exceed 10% of the coating thickness. Therefore, any influence from the underlying substrate when indenting the Si–N–Fe–C coatings is unlikely. Silicon is also known to undergo a phase transformation [52,53,54,55] when exposed to load. This process was thoroughly investigated with nanoindentation. At low loads, the phase transformation can be observed as a large degree of depth recovery during the un-loading curve [52,54,55]. This behavior can be seen for the indentations on uncoated Si wafers, but not for the coated wafers, as can been seen in Appendix A.

#### 4.3.1. Hardness

The nanoindentation investigations showed a clear difference in hardness (13–17 GPa) and modulus (190–212 GPa) between the different investigated points. Previous SiN_x_ coatings exhibited a hardness of between 12–26 GPa [17,23,24]. Earlier studies have also looked at the incorporation of carbon, which yields a hardness lower than that of SiN_x_ coatings. The study by Pettersson et al. showed a maximum hardness of 21 GPa for SiN_x_ coatings, while coatings that contained carbon had a slightly lower hardness (10–19 GPa) [34]. Carbon-containing SiN_x_ coatings were also studied by Berlind et al., who obtained a hardness ranging between 9–28 GPa [16], i.e., also encompassing the values found in this study.

A difference in hardness between the different points was found, with correlations to silicon (*p* < 0.05), iron (*p* < 0.05), nitrogen (*p* < 0.05) and oxygen (*p* < 0.05). Indeed, as silicon increases, so does nitrogen, and hence the material is expected to behave more like a ceramic for high silicon and nitrogen content, and also have a higher hardness compared to the points with high iron content (points 2 and 4). 

#### 4.3.2. Indentation Modulus

Earlier studies looked at the Young’s modulus of the coatings and found a wide range between approximately 170 GPa and 290 GPa for SiN_x_ coatings [23] and 140 GPa and 220 GPa for SiNC coatings [34]. If a Poisson’s ratio of 0.25 is assumed, as was used in previous work with SiN_x_ coatings [23,24], the Si–N–Fe–C coatings would have a Young’s modulus of 178 to 199 GPa, which is well within the range found for similar coatings.

As with the hardness, the modulus correlates to the x-coordinate (*p* < 0.05). In addition, the modulus correlates to silicon (*p* < 0.05), iron (*p* < 0.05) and nitrogen (*p* < 0.05). As silicon increases so does the modulus, while the opposite effect can be observed for iron. This is likely due to the increasing ceramic behavior in points 1 and 3, which contain more silicon and nitrogen compared to points 2 and 4. Furthermore, there is a correlation between the modulus and oxygen (*p* < 0.05). As oxygen decreases so does the modulus, which can be explained by the simultaneous decrease of nitrogen that could make the material more metallic.

### 4.4. Cell Response

An osteoprogenitor cell line (MC3T3) was used to evaluate the biocompatibility of the coatings in a first test. The cells adhered to the surface and displayed a similar morphology as compared to the positive control, which in this case was cells seeded on a tissue culture plastic petri dish. The cells spread across the test sample and the positive control with a comparable variation, making any further conclusions on the difference between the points impossible with this type of in vitro set-up. Still, the cells demonstrated a comparable response to the material, with cells spreading and dividing across the test sample, indicating a similar initial level of biocompatibility as the positive control. There are studies suggesting iron has a cytotoxic effect [56,57] but that was not confirmed in this study. This is in accordance with previous studies on the biocompatibility of silicon nitride, where in vitro studies found osteoblask-like cells to grown and proliferate on polished silicon nitride discs [29] as well as silicon carbon nitride coatings [58] and no cytotoxic effects when exposing silicon nitride material to fibroblast cells [30]. In addition, Si_3_N_4_ was found to have superior osseintegration properties in vivo in rats [28], which further suggest the potential of silicon nitride as a material for joint implants.

### 4.5. Limitations of the Study

When working with gradients it is important to minimize the area from which information is obtained, and depending on analysis technique, the size of the area might differ. The results in this study are based on a largest average area of approximately 2 mm × 2 mm.

The MC3T3 cells were seeded over the entire wafer and the cells might have been able to migrate over the sample and result in skewed results. Had one composition been favorable for cells it is possible that these cells would migrate into areas with less favorable composition. Also, a higher number of samples may give statistical significance for these tests of naturally high variation.

### 4.6. Future Work

The influence of composition on the adhesion has not been investigated in this study as it is a system parameter and needs to be investigated for the coating on the intended implant substrate material. This will be investigated in future studies with suitable substrates. 

The wear properties were not reported, as the coatings in this study consist of gradients in both x and y directions, making it impossible to investigate a wear track with a consistent composition. It is also important to use a relevant substrate material as it may influence the wear performance of the coating. Therefore, wear properties should be investigated on homogenous coatings on relevant substrates.

The dissolution in aqueous solutions and the biocompatibility over time also needs investigation.

## 5. Conclusions

The use of combinatorial sputtering proved to be a useful technique in generating a wide range of compositions in a single sample silicon (26 to 34 at.%), iron (10 to 20 at.%), carbon (8 to 14 at.%) and nitrogen (40 to 47 at.%). The composition had a clear influence on the mechanical properties with H ranging from 13.7 ± 0.9 GPa to 17.3 ± 0.7 GPa and M from 190 ± 7 GPa to 212 ± 8 GPa, both revealing a positive correlation to silicon and nitrogen and the opposite for iron. These results, in combination with a consistently low surface roughness (approximately 2 nm), amorphous crystal structure and biocompatibility make the coatings a potentially interesting alternative for orthopedic implants. Dissolution properties remain to be investigated as well as the effect that this would have on cells. This could be done by isolating small areas on the samples to obtain extracts from regions with a relatively consistent composition. The extract can then be used to evaluate what elements are leached out as well as the effect this has on cells in an in vitro study. 

## Figures and Tables

**Figure 1 materials-13-02074-f001:**
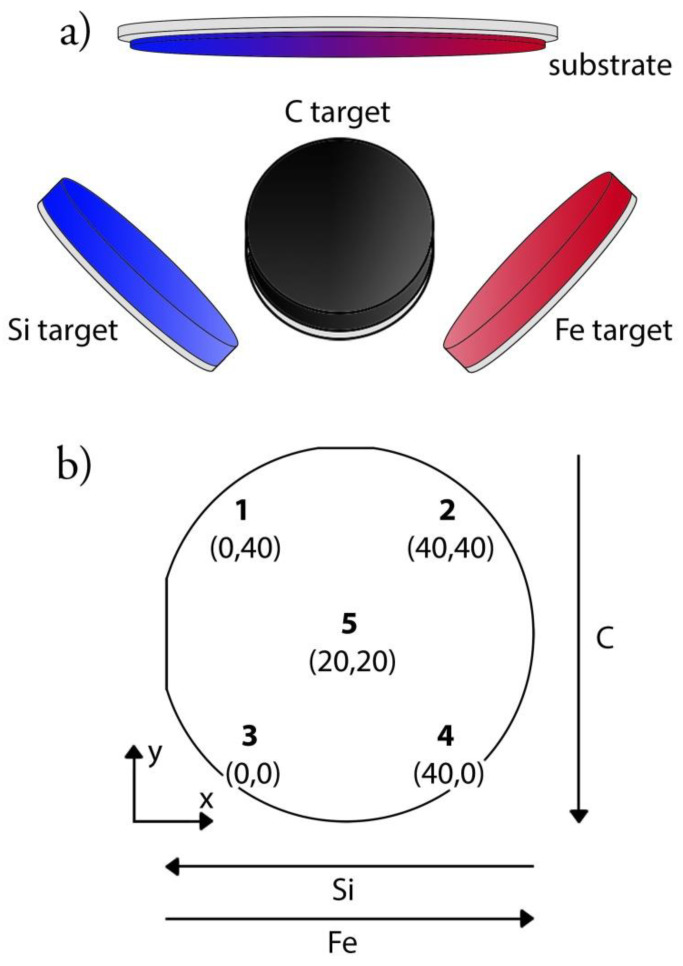
Deposition setup displaying (**a**) the three targets (silicon, iron, and carbon) and the position of the silicon wafer substrate, and (**b**) a schematic image of the sample with the five investigated points and their coordinates in mm with point 3 as origin.

**Figure 2 materials-13-02074-f002:**
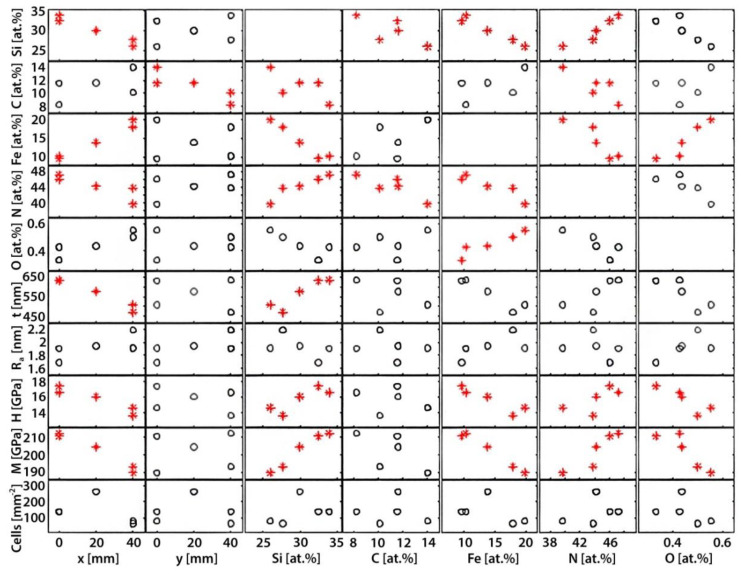
An overview of the various input and output parameters and their correlations. Each rectangle represents a graph where the input parameters are found on the x-axis and the output on the y-axis. Please note that the compositional elements (Si, C, Fe, N and O) are both output and input parameters as they vary as a function of the coordinates (x and y) and are important to consider as input parameters for the thickness, surface roughness (R_a_), mechanical properties (H and M), and cell density. Cases where there was a statistically significant correlation were marked with red stars rather than black circles.

**Figure 3 materials-13-02074-f003:**
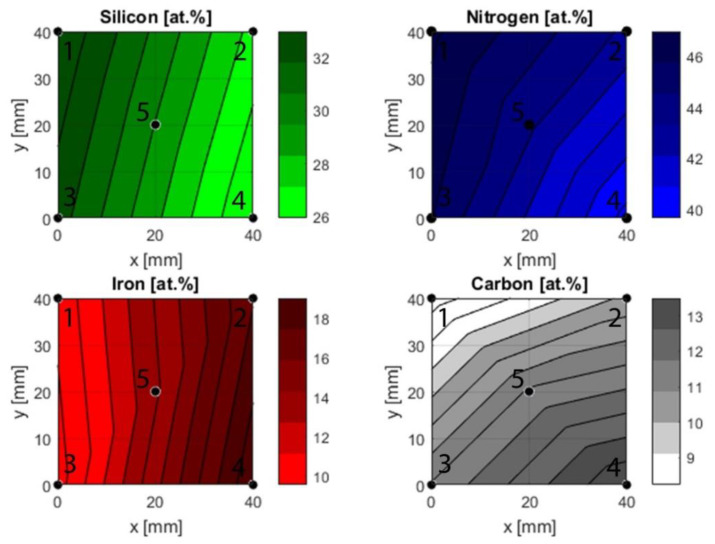
Compositional gradients based on ToF-ERDA measurements. The fit is created using a piecewise linear interpolation that fits a linear polynomial between sets of three points, e.g., 1, 3 and 5.

**Figure 4 materials-13-02074-f004:**
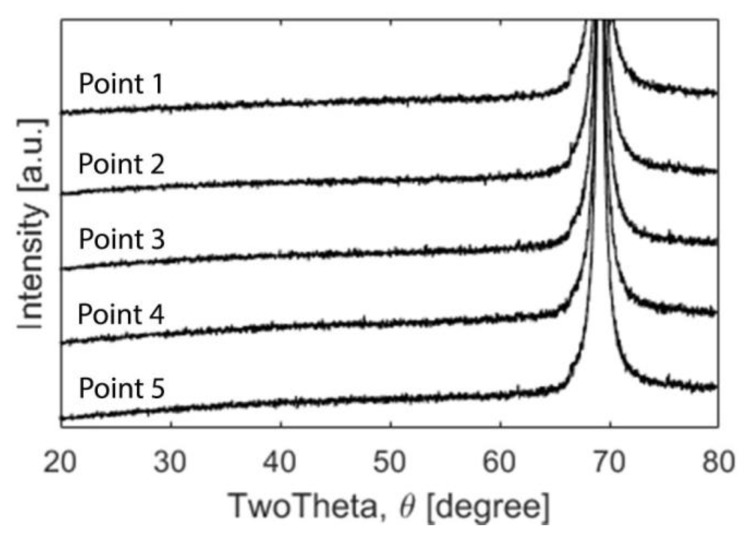
XRD diffractograms for points 1–5. The only detectable peak was the Si (400) at 69°.

**Figure 5 materials-13-02074-f005:**
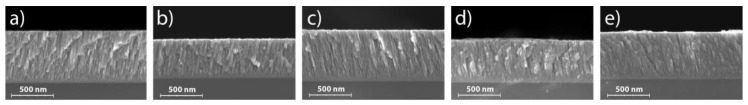
SEM images of a fractured cross-section and the surface for (**a**) point 1, (**b**) point 2, (**c**) point 3, (**d**) point 4 and (**e**) point 5. The images reveal a columnar structure and differences in thickness for all five points.

**Figure 6 materials-13-02074-f006:**
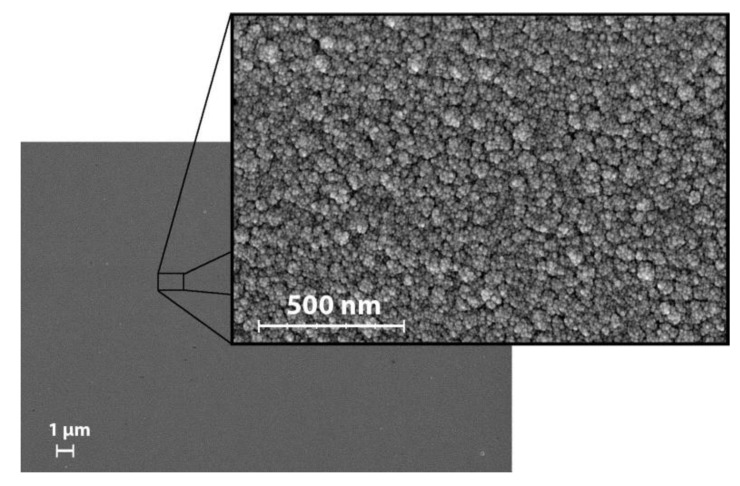
SEM images of the surface in point 1. All points exhibited a similar appearance.

**Figure 7 materials-13-02074-f007:**
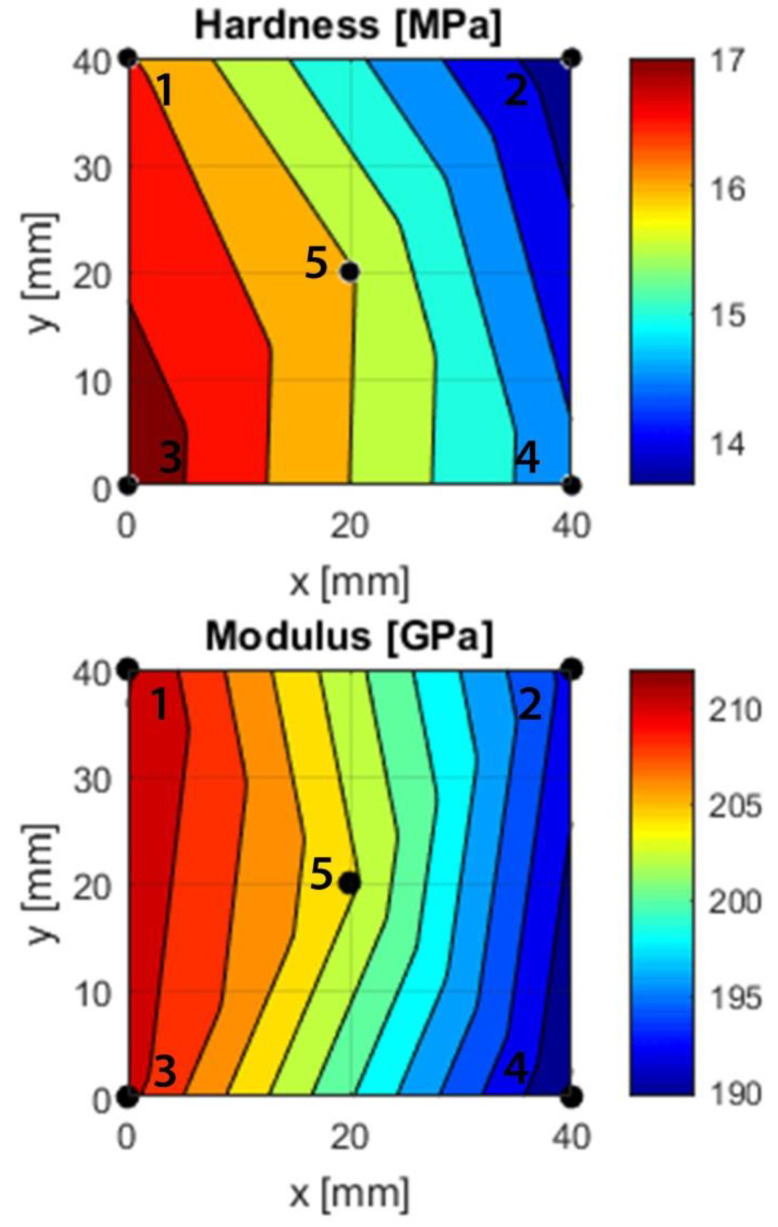
Hardness and modulus gradients based on nanoindentation measurements at points 1–5. A piecewise linear interpolation between sets of three points was used to create the fit. The modulus relates to the Young’s modulus by (1 − ν^2^).

**Figure 8 materials-13-02074-f008:**
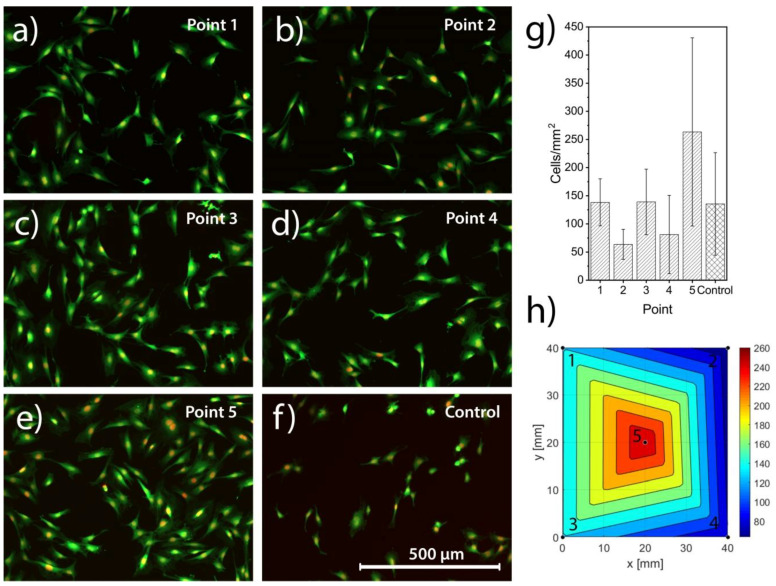
MC3T3 cells stained with Hoechst and CFDA-SE and imaged in a fluorescent microscope for point 1 in (**a**), point 2 in (**b**), point 3 in (**c**), point 4 in (**d**), point 5 in (**e**) and the tissue culture plastic control in (**f**). The quantified number of cells/mm^2^, shown in (**g**), displays high standard deviations and no significant differences, neither between points nor to the control. To better visualize the results the cell density is presented as a color plot, (**h**).

**Table 1 materials-13-02074-t001:** The obtained composition (Si, Fe, C, N and O), thickness, surface roughness, mechanical properties (H and M) and cell density.

Point	x [mm]	y [mm]	Si [at.%]	Fe [at.%]	C [at.%]	N [at.%]	O [at.%]	t [nm]	R_a_ [nm]	H [GPa]	M [GPa]	Cell Density [cell mm^−2^]
**1**	0	40	33.9 ± 0.5	10.2 ± 0.3	8.2 ± 0.1	47.2 ± 0.7	0.4 ± 0.1	633	1.9 ± 0.5	16.5 ± 1.2	212.1 ± 8.0	138 ± 42
**2**	40	40	27.6 ± 2.8	18.2 ± 3.4	10.0 ± 1.5	43.7 ± 0.9	0.5 ± 0.1	467	2.2 ± 0.3	13.7 ± 0.9	193.2 ± 6.9	63 ± 27
**3**	0	0	32.5 ± 0.4	9.6 ± 0.4	11.6 ± 0.3	46.0 ± 0.8	0.3 ± 0.2	630	1.7 ± 0.1	17.3 ± 0.7	210.6 ± 5.8	139 ± 58
**4**	40	0	26.0 ± 0.9	20.0 ± 0.3	13.9 ± 0.2	39.7 ± 0.7	0.6 ± 0.03	507	1.9 ± 0.1	14.7 ± 0.8	189.8 ± 6.9	81 ± 70
**5**	20	20	30.0 ± 0.1	13.8 ± 0.4	11.6 ± 0.2	44.2 ± 0.4	0.4 ± 0.1	576	1.9 ± 0.1	16.0 ± 1.1	204.4 ± 7.8	263 ± 167

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
