# Peer review of "Si–Fe–C–N Coatings for Biomedical Applications: A Combinatorial Approach"

_materials, 2020, doi:10.3390/ma13092074_

Round 1

Reviewer 1 Report

The manuscript reports an investigation on the properties of coatings with compositional gradients. In detail, Si-Fe-C-N coatings, obtained with combinatorial sputtering, were prepared and analyzed in terms of composition, surface and mechanical properties, and cells adhesion with the aim to highlight their potential application in the field of orthopaedic implants. The aim of the study is of interest, since the potential benefits of using ceramic coatings for biomedical applications are manifold; the methodology to produce the coatings was effective but the results obtained in relation with the proposed application are very preliminary and further investigation is required. Some detailed comments are reported here below.

Detailed comments

Lines 30-31: The sentence seems to report a prevision for a future trend, but past years are cited. Please check.

Lines 111-112: Can these systematic uncertainties affect the measurements performed on the coatings, since the percentage of carbon found was between 8 at.% to 14 at.%?

Line 113: A brief description of the mentioned software should be given.

Line 126: In the title of the subparagraph should be specified that the indentation modulus was evaluated.

Line 129: A number at the beginning of the sentence should be spelled out as word.

Lines 159-160: ''..  is also presented in colour- and bar plots.''. These diagrams seem to be missed. However, if the Authors refer to the following iso-colour representations, for clarity, it should be stated here. Please check.

Line 164: The representation of the results reported in Figure 2 is not clear. It is suggested to consider the possibility to choose a more significant graphical representation, or split the content in more figures, or try to improve the reported representation to better highlight the statistical significant correlations.

Line 199: The sentence is not clear. Please try to better explain the content with a more extensive sentence  or more sentences.

Lines 231-233: For clarity, the content of the diagrams reported in Figure 8g) and 8f) should be explained in the text.

Lines 245-246: The Authors have related the measured mechanical properties in the different areas of the coating. At what extent the variation of thickness can be relevant for these properties?

Lines 390-397: Conclusions should be extended describing more in detail future research on the topic.

Reviewer 2 Report

In the present paper, the authors developed a new method to deposit a big range of Si-De-C-N coatings with different compositions. Also, they studied the mechanical propertoes, surface morphology and crystal structure, as well as the biocompatibility. The authors claimed that this method and the generated materials are of interest for biomedicine.

This is an overall interesing and multicisciplinary article which offers some new perspectives. The manuscript is generally well-written and they have done a good job generating conclusiones consistent with the results. Thus, I suggest acceptance in present form.

Author Response

In the present paper, the authors developed a new method to deposit a big range of Si-De-C-N coatings with different compositions. Also, they studied the mechanical propertoes, surface morphology and crystal structure, as well as the biocompatibility. The authors claimed that this method and the generated materials are of interest for biomedicine.

This is an overall interesing and multicisciplinary article which offers some new perspectives. The manuscript is generally well-written and they have done a good job generating conclusiones consistent with the results. Thus, I suggest acceptance in present form.

The authors thank the reviewer for the time and effort spent on reviewing the manuscript and appreciate the positive feed-back.

Reviewer 3 Report

Review on the manuscript “Si-Fe-C-N coatings for biomedical applications: a combinatorial approach”

by

Charlotte Skjöldebrand, Gry Hulsart-Billström, Håkan Engqvist and Cecilia Persson

The authors present an interesting paper entitled “Si-Fe-C-N coatings for biomedical applications: a combinatorial approach”. The results are valuable, sound, and original. The paper is clearly presented and well organized. All illustrations and tables are necessary and adequate. The article contains the data of scientific researches on the modes of depositions, structure and properties of complex Si-Fe-C-N films. The method of the reactive magnetron sputtering with combination of the sputtered targets was used for deposition of gradient coating on the surface of the single sample. This principle is widely used for deposition of the composite coatings and the article could be of interest for the researchers in the field of PVD. However, there are some points that must be clarified before publishing and some minor improvements are suggested to increase the scientific value of the paper.

  1. The authors declare (lines 75-77) that the aim of the study is “…to investigate the mechanical properties, surface morphology, crystal structure and biocompatibility of combinatorially sputtered silicon nitride coatings incorporating carbon and iron.”

But the conclusions say (lines 390, 391):   “The use of combinatorial sputtering proved to be a useful technique in generating a wide range of compositions in a single sample…”. So, the questions arise: 1) Is the goal of the study achieved? 2) Phase composition of the coating and its structure remain unclear?  3) Why silicon nitride? 4) Are biocompatibility data available? 5) May be it is better to reformulate the aims of the research?

  1. Line 93. “The gas flows were 3 sccm and 10 sccm respectively, giving a nitrogen-to-argon gas flow ratio (N2/Ar) of 0.323.” What is the reason for choosing of this ratio N2/Ar?
  2. Lines 98, 99. “During deposition, the substrate was heated to a temperature of 350˚C.” Why 350°C? How the substrate was heated and how the temperature was measured?
  3. Is the structure of the coating depends on the substrate temperature?
  4. How the operating mode of the installation was controlled?
  5. How stable did the installation is worked?
  6. Deposition time is not indicated and silicon wafer diameter is not specified.
  7. Each value presented in Table 1 is characteristic for local position 1, 2,3,4,5. What about the positions (eg.) (10,20) or (30.30) or another? Is there any reason to think that any parameter changes depending on the coordinates as shown in Fig. 3 and Fig. 7.
  8. It is known that the thickness and parameters of the film depend on the power of the magnetron discharge. Plasma density is heterogeneous and depends largely on the configuration of the targets erosion zones. What can be said about the power density and its distribution on the substrate? How the presented distributions are related to the distribution of power density on the condensation surface?

I recommend the publication of this work after minor revision.

Reviewer 4 Report

Commented manuscript is attached in the review form.

Manuscript titled "Si-Fe-C-N coatings for biomedical applications: a combinatorial approach" is well-written and highly logical. Findings are clearly stated and methods that allowed the authors to reach these conclusions are well described. I have only a few minor problems with the manuscript, but these must not stop the publication. Therefore, I accept with minor revision.

Reviewer 5 Report

This is an interesting paper and worth publishing in my opinion. It is well written and the results are thoroughly discussed. I only have a couple of remarks:

  • Page 3, line 92: Consider replacing "Torr" and "sccm" with the appropriate SI units.
  • Section 2.2.2: What type of stage and sample holder were used? I think this is some important information that should be included, since your samples appeared to be XRD amorphous.
  • Section 2.2.1: You state that the surface roughness was analyzed over an area of 5x5 µm2. Did you estimate Sa (areal roughness parameter) or only Ra (as in Table 1)?
  • Section 2.2.3: For the reader, you should define here that in the following "H" and "M" were used for hardness and indentation modulus, respectively. Otherwise these abbreviations may lead to confusion.
